# OpenReview forum: "Sampling-aware Adversarial Attacks Against Large Language Models"
_ICLR.cc/2026/Conference — ICLR 2026 Poster_

### Official Review · Reviewer_PvN7 · 2025-10-28

**Soundness:** 3
**Presentation:** 3
**Contribution:** 3
**Rating:** 6
**Confidence:** 3

**Summary:**

This paper proposes a sampling-aware framework for adversarial attacks against large language models (LLMs). The key idea is to integrate sampling explicitly into the optimization process, treating attack design as a compute-budget allocation problem between optimization and sampling. The authors show that repeated sampling during attack iterations can improve attack success rate (ASR) and efficiency, introducing a label-free entropy-maximization objective as a proof-of-concept.

**Strengths:**

[1] The paper correctly identifies that most current attacks evaluate only greedy generations and ignore stochastic sampling, which can underestimate real-world risk.
[2] Experiments include multiple baselines (GCG, PAIR, BEAST, AutoDAN, etc.) and diverse open-weight models.
[3] The entropy-maximization loss is interesting as a model-agnostic, label-free alternative that naturally leverages sampling variance.

**Weaknesses:**

[1] The paper claims an “optimal trade-off” between optimization and sampling but provides no analytical derivation or formal proof.
[2] The harm function h(y) is mentioned but not precisely defined; the dependency on the judge model (StrongReject) is underexplained.
[3] Only sub-10B open-weight models are used; conclusions about “LLM safety at scale” are overgeneralized.
[4] It maximizes first-token entropy only; rationale for focusing on the first token rather than full sequence entropy is missing. A short experiment or discussion on multi-token entropy or conditional entropy over first k tokens would strengthen Section 6.4.
[5] The categories “refusal,” “incomplete,” and “harmful” appear visually distinct, yet thresholds (0.1, 0.3, 0.5) seem arbitrary. Adding a short justification or sensitivity test for these cut-offs would substantiate the observation.

**Questions:**

[1] How sensitive are the results to the choice of harm threshold 0.5? Would conclusions change under stricter thresholds?
[2] The three sampling schedules (optimize-then-sample, uniform, block sampling) are reasonable, but their selection appears heuristic. It would strengthen the argument if the authors could explain why these three were chosen, or at least provide intuition on when each schedule is preferable.
[3] The entropy objective only considers the first-token distribution, which may not capture downstream harmful semantics. It would be helpful to discuss why entropy is computed only at the first token and whether extending it to later tokens affects stability or optimization cost.
[4] Because the main claim concerns stochastic sampling, omitting temperature and top-k values leaves the setup ambiguous. A table summarizing decoding hyperparameters for all models would remove this uncertainty.

---

> ### Author Response · Authors · 2025-11-19
> **Answer to PvN7**
>
> We thank you for the helpful review and for recognizing the real-world relevance of our work.
> Regarding the mentioned concerns:
>
> ### W1 Lack of derivation for optimal trade-off
> We empirically determine optimal compute trade-offs for fixed sampling schedules like *optimize-then-sample*. The resulting compute-optimal frontiers trace out empirical optima and can be displayed as in Figure 3. As described in L218-220, a closed-form solution does not appear tractable in the general case, so we provide an empirical characterization instead.\
> We also added a more precise explanation and mathematical definition of the compute-optimal frontier to App. M.
> ### W2 Definition of h(y) and dependence on StrongREJECT
> We introduce the harm function in Section 3, where we define it as a function $h$ mapping from the space of completions $V^*$ to a normalized harmfulness score in [0, 1]. We instantiate such a function via the StrongREJECT judge.\
> We added new results from a study with **human judgments** to Appendix K (excerpt below), showing that **correcting for false positives of the StrongREJECT model does not alter our takeaways**.\
> Please also see the general comment for a more thorough discussion of the judging approach and additional data.
> |Metric|AutoDAN|BEAST|PAIR|REINFORCE-GCG|GCG|
> |-|-|-|-|-|-|
> |ASR$_q@1$|0.34|0.08|0.08|0.23|0.17|
> |ASR$_q@1$ human|0.27|0.06|0.06|0.18|0.13|
> |ASR$_q@50$|0.51|0.22|0.24|0.55|0.47|
> |ASR$_q@50$ human|0.50|0.21|0.23|0.53|0.45|
> |Δ$_{1\rightarrow50}$|0.18|0.14|0.16|0.32|0.30|
> |Δ$_{1\rightarrow50}$ human|**0.24**|**0.15**|**0.17**|**0.35**|**0.32**|
>
> ### W3 Claims about safety at scale
> We apologize for the confusion - our comments regarding "safety at scale" refer to _deployments_ at scale, i.e., models which can be accessed by a large number of users or are hit by a large volume of requests, not parameter count. We have clarified this more in the revised paper.\
> We were nonetheless interested in how our approach performs on larger models and added experiments on Llama 3.1 70B to App. N, confirming that our method works at larger parameter scales too.
>
> ### W4/Q3 Rationale for first-token entropy
> We conducted preliminary experiments with full sequence entropy and other alternatives; however, we selected the first-token formulation for three reasons:
> 1. Alternatives had similar performance (results below from early ablation) and (slightly) higher compute cost.
> 2. Full sequence entropy more often led to generations diverging or becoming incoherent.
> 3. First-token entropy is a far more practical objective for black-box settings where it can be relatively efficiently computed using only last-token logit access. In comparison, full sequence or k-token entropy is prohibitively expensive to compute in this setting.\
> To clarify our reasoning, we have added this discussion to the revised version of the paper.
> |Model|Full-Seq ASR@1|Full-Seq ASR@50|First-Token ASR@1|First-Token ASR@50|
> |-|-|-|-|-|
> |Llama 3 8B CB|0.01|0.04|0.01|**0.07**|
> |Llama 2 7B DA|0.03|0.29|0.03|**0.30**|
> |Gemma 3 1B|0.07|**0.56**|0.06|0.51|
> |Llama 3.1 8B|0.26|**0.73**|0.21|0.69|
> ### W5/Q1 Sensitivity to thresholds
> **Sampling-aware attacks remain effective across various harm thresholds**; in fact, many of our results do not rely on thresholding at all and instead directly use the unthresholded expected harmfulness $\mathcal{H}$, for example in Figures 2-4, 20-28, 31, and 32.\
> We have updated Figure 8 to show results across different thresholds (including stricter ones), demonstrating strong insensitivity to the exact values chosen.\
> We would also like to clarify that the value 0.3 is only used to describe the bi/tri-modal shape of the harmfulness histograms and is not used in any experiments.\
> To further evaluate the sensitivity of our results to the judge model and chosen threshold, we conducted a human study (which also does not require thresholding) and include results in Appendix K, showing that the key findings of our work also appear when using gold-standard human labels.
>
> ### Q2 Explanation and intuition for the three sampling schedules
> We expanded on the rationale for selecting the three heuristic schedules in the revised Section 5.1. Schedule selection involves two key intuitive trade-offs: 1) sampling later is likely to be more effective as the underlying prompt has been optimized for longer, and 2) spreading sampling across multiple prompts is likely to be better than sampling from a single prompt since it introduces more diversity.
> The schedules are designed to explore different configurations of these basic trade-offs.
> ### Q3 Why is entropy computed only at the first token & does extending it to later tokens affect stability or optimization cost?
> Please see W4.
> ### Q4 Decoding hyperparameters
> We use temperature 0.7 with multinomial sampling in our experiments. In response to your comment, we have added a precise decoding parameters to Section 6.1 and App. L.
>
> We hope our comments have addressed your questions!

---

> > ### Comment · Reviewer_PvN7 · 2025-11-24
> >
> > Well received the answers and feedback. Thank you very much! Will keep the score.

---

### Official Review · Reviewer_gbXv · 2025-10-30

**Soundness:** 2
**Presentation:** 2
**Contribution:** 2
**Rating:** 4
**Confidence:** 4

**Summary:**

The paper proposes a sampling-aware framework for adversarial attacks against LLMs that treats repeated sampling as a core and efficient part of the attack itself. It reframes the attack as a "resource allocation problem" where an attacker must decide how to spend a fixed computational budget: either on optimization or on sampling. The paper proposes three sampling schedules to explore this trade-off: optimize-then-sample, uniform sampling, and block sampling. Additionally, the paper introduces a label-free attack objective based on maximizing the entropy of the model's first token. During sampling-aware optimization, the framework generates n samples from the optimized query in each interaction. Finally, it selects the query that elicits the highest ASR (i.e., the largest number or proportion of malicious samples) as the final optimized query. The paper evaluates the proposed method on open-source LLMs, such as Gemma-3-1B, Llama-3.1 8B, Llama-3-8B-CB, and Llama-2-7B-DA. It samples 100 prompts from HarmBench for the experiments. The evaluation results show that it can improve the ASR by up to 37%.

**Strengths:**

1. The proposed framework can be integrated into the existing optimization-based attack methods, such as GCG, AutoDAN, and BEAST.
2. The paper introduces a label-free objective, which is interesting.
3. The paper frames the adversarial attack against LLMs as a resource allocation problem, which provides a fresh perspective for this field.

**Weaknesses:**

1. **Robustness of StrongREJECT.** The ASR is calculated by StrongREJECT, which is a judge model designed for detecting malicious content. Although this judge model is designed to reduce the possibility of false positives, it still cannot achieve 100% accuracy; it may still have false negatives or false positives. Especially when comparing the results of ASR@50 (generating 50 samples) and ASR@1 (generating one sample), this effect could be amplified. The results would be more reliable if the paper can randomly sample from the generated results and conduct human evaluation to compare the inter-annotator agreement between humans and the judge model. This can assess the reliability of the judge. Secondly, HarmBench also provides a judge model; maybe the paper can also provide results using that judge to see whether the results are aligned.
2. **Evaluation on larger models.** Based on the cost calculation, the low sampling cost appears to stem from using small models (1B–8B). However, these small LLMs may have substantially lower utility than larger models (e.g., GPT, Gemini, Claude). For attacks against larger LLMs, it is unclear whether the sampling would remain cheaper than optimization. Some optimization-based attack methods can optimize on small-size LLMs and then transfer to larger LLMs. It is suggested to report results on larger LLMs (e.g., Llama-70B) to clarify this.
3. **Unclear result.** It is unclear which optimization strategy is used for the reported results in evaluation (e.g., optimize-then-sample).
4. **Comparison with other sampling-based methods.** One main takeaway from this work is that sampling can also contribute to ASR, and considering the trade-off between sampling and optimization can elicit the highest ASR. However, this concept is not new, and prior work has proposed a sampling-based attack method [1]. The paper should compare the proposed method with prior work [1] to demonstrate its advantage.

[1] Huang et al., Catastrophic Jailbreak of Open-source LLMs via Exploiting Generation, ICLR’24

**Questions:**

1. According to Figure 7, the ASR appears to plateau (stop increasing) very early, around 20% of the total iterations. Why does the paper not use early stopping to reduce cost?
2. Under a fixed budget, if the attack invests most of the budget in sampling (e.g., 50 samples per iteration for 5 iterations), how does transferability change? Can the attack still demonstrate strong transferability compared to the original setting (sampling only once at the end of optimization)?
3. How to find the right balance? It seems that sampling more in each iteration could lead to a higher ASR. But how many samples should be drawn? Is there any guidance on this?
4. Does the ASR reported in Figure 1 correspond to ASR@50 or ASR@1?

---

> ### Author Response · Authors · 2025-11-19
> **Answer to gbXv**
>
> We are excited to hear that you appreciated our fresh perspective and found our novel objective interesting.
> We would like to address your concerns below:
> ### W1 Ask for human judgment or alternative judges.
> While no judge is 100% accurate, our evaluations of open-source judge models (including HarmBench, Llama Guard 3, and StrongREJECT) showed that StrongREJECT was the strictest judge that also aligned best with humans. In contrast, alternatives like HarmBench produced significantly more false positives and near-binary outputs that prevent more nuanced analysis.
>
> In response to your comment, we added a human study to Appendix K, characterizing the impact of imperfect automated judging. We find that while false-positive correction reduces ASR in all settings, our conclusions about sampling-aware attacks remain unchanged, in particular about the associated efficiency and performance advantages.
> A counterintuitive finding is that ASR@50 is _less_ affected by false positive corrections than ASR@1. The reason is that the vast majority of successful attacks include multiple judged-harmful completions among the set of $n$ - the probability that all of them are false positives is low.
>
> Please also see the general comment for more data and discussion.
> |Metric|AutoDAN|BEAST|PAIR|REINFORCE-GCG|GCG|
> |-|-|-|-|-|-|
> |ASR$_q@1$|0.34|0.08|0.08|0.23|0.17|
> |ASR$_q@1$ human|0.27|0.06|0.06|0.18|0.13|
> |ASR$_q@50$|0.5|0.22|0.24|0.55|0.47|
> |ASR$_q@50$ human|0.50|0.21|0.23|0.53|0.45|
> |Δ$_{1\rightarrow50}$|0.18|0.14|0.16|0.32|0.30|
> |Δ$_{1\rightarrow50}$ human|**0.24**|**0.15**|**0.17**|**0.35**|**0.32**|
> ### W2 Evaluation on larger models
> We added experiments using BEAST and PAIR vs Llama 3.1 70B to Appendix N, finding that **performance improvements from sampling also apply to larger models**. Results for AutoDAN and GCG are more compute-intensive and will follow in the camera-ready.
> ### W3 Which optimization strategy is used for the results?
> We use optimize-then-sample unless reported otherwise and chose this default due to its simplicity and good performance. We have clarified this in the revised paper.
> ### W4 Comparison with [1]
> A key difference between [1] and our work is that we combined sampling with adversarial attacks, whereas [1] uses only sampling without any adversarial attack or optimization component. This difference allows us to achieve much higher success rates, as evidenced by new results in App. J, which compare the use of different generation strategies setting similar to [1] (w/o adversarial attacks).\
> These experiments show that pure generation is not able to reach the same harmfulness as, e.g., sampling-aware GCG - especially on robust models. For example, for Gemma 3, we achieve an average harmfulness of 0.87 at $n=50$, but a setting similar to [1], even with $n=1000$, is still significantly lower at 0.51. This finding is consistent across the other models as well (Llama 2 7B DA 0.52@50 vs 0.09@1000, Llama 3.1 8B 0.79@50 vs 0.56@100, and Llama 3.1 8B CB 0.11@50 vs 0.03@100).
>
> [1] Huang, Yangsibo, et al. "Catastrophic jailbreak of open-source llms via exploiting generation." arXiv:2310.06987 (2023).
> ### Q1 Early stopping
> We use early stopping in our Pareto analyses, where we find that taking fewer optimization steps combined with more sampling yields higher ASR. However, some sampling schedules, like the uniform schedule with few per-step samples, require taking a larger number of steps. Optimizing for longer also generates multiple prompt candidates, which we conjectured to be more effective to sample from vs multiple samples from the same prompt.
> ### Q2 Transferability of attack prompts
> Most studied attacks (except GCG) do not claim transferability in their original setup. As we leave the underlying optimization untouched, we see no reason the optimized prompts should transfer worse than the baseline. However, if the optimization is stopped early and combined with a large amount of sampling to retain ASR, we expect the underlying prompt to be somewhat weaker, both on the 'source' and transfer 'target' model.
> ### Q3 Guidance on how many samples to draw
> Great question! Figure 3 and L328-340 aim to address this. The key finding is that when aiming to elicit maximum harm, **more sampling (and less optimization) is better up to at least 500 samples** (the maximum we explored in our experiments) and likely even more. If one instead just wants to **match the harm of the greedy baseline with minimum cost, around 100 samples are ideal**.
> Across **all models and attacks, the greedy baseline is suboptimal** and simply using 10 samples (usually more) would always improve ASR.
> ### Q4 ASR in Figure 1
> Figure 1 shows ASR$_b$ for the optimize-then-sample schedule, meaning ASR at constant FLOPs budget, without a specific query count associated with it. As described in the previous answer, typically, at a given budget, sampling more and optimizing less leads to maximized ASR.
>
> We hope these answers address your concerns!

---

### Official Review · Reviewer_uZGG · 2025-10-30

**Soundness:** 2
**Presentation:** 3
**Contribution:** 2
**Rating:** 6
**Confidence:** 3

**Summary:**

This paper studies the worst-case adversarial attack against LLMs. Existing methods focus on optimizing adversarial tokens to achieve a high average ASR across samples, whereas this paper focuses on the worst-case ASR across all samples. That is, although the adversarial token may not consistently trigger the harmful response, as long as it can trigger it once, the attack is considered successful. Therefore, the authors propose conducting additional sampling to identify rare yet harmful cases. Experimental results show that with more sampling, existing adversarial attacks can improve in the worst-case scenario. The authors also propose an objective to increase the randomness of the generated responses to reach more samples and thus trigger the adversarial effect.

**Strengths:**

1. This paper studies the sampling aspect, which was neglected in many existing methods. This is an interesting aspect.

2. With more sampling budget, existing methods can also be improved.

3. Even without any target response or an affirmative target template, the label-free entropy-maximization objective can trigger harmful responses.

**Weaknesses:**

1. There seems to be a mismatch between the design and the evaluation. For example, Algorithm 1 involves the dynamic interaction between the optimization and sampling processes. However, during the evaluation, the prompts and samples are stored to explore the different sampling schedules post-hoc. This may not reflect the real performance and is less supportive.

2. Only StrongReject is used as the harm score measurement.

3. Although Figure 8 shows that the Frequency of (h(y)>0.5 | not refusal) doesn't change, this can not lead to the conclusion that GCG didn't shift the harmfulness of the compliant responses. For example, 10 out of 20 Ys of a harmful score of 0.8 is different from 10 out of 20 Ys of a harmful score of 0.6. Maybe draw the lines with different thresholds or a histogram of the scores.

4. It's unclear what temperature is used for sampling. More randomness can also be achieved by increasing the temperature, besides the high entropy objective.

5. Since this attack relies on sampling, it's unclear if the harmful response or behavior can be consistently reproduced.

1. Why isn't there a yellow bar (250 samples) for the optimize-then-sample case in Figure 4?

2. How to understand the compute-optimal frontier in Figure 3? If the total FLOPS budget is fixed, then increasing the optimization FLOPS should decrease the sampling FLOPS.

**Questions:**

1. In addition to the StrongReject, will other judges, such as Llama-Guard or GPT-4-Judge, give similar and consistent results?

2. If we don't use any loss objective, but use a higher temperature, will it achieve a similar adversarial worst-case effect?

---

> ### Author Response · Authors · 2025-11-19
> **Answer to uZGG**
>
> We thank the reviewer for the detailed review and are glad they found our findings interesting.
> ### W1 Lack of interaction between optimization and sampling
> We agree that methods that dynamically interleave optimization and sampling are an interesting direction. However, in the context of this paper, we restricted our focus to two goals:
>
> 1) In Algorithm 1, we introduce a framework to consistently evaluate attacks with sampling.  To be applicable both to existing attacks (which generally do not leverage dynamic interactions between sampling and optimization) and future ones that may follow a different paradigm, it is **designed to be very flexible and general**.
> 2) To show that this framework is meaningful, we present an **empirical evaluation with existing attacks cast in this framework** and show that they already benefit from this perspective.
>
> We hope our framework can serve as a foundation for future work that e.g., more thoroughly explores dynamic sampling/optimization interactions. We clarified this scope in the revision.
> ### W2/Q1 Alternatives to StrongREJECT
> We added a study using human judging to Appendix K, showing that **our key findings are not sensitive to a particular classifier**. The humans followed standard instructions from [1] and were blinded to model, attack, and generation parameters.\
> We selected StrongREJECT after thoroughly evaluating it against other open-source judges, including LlamaGuard 3 and HarmBench's classifier, finding it to be the strictest judge with the lowest false positive rate. In addition, other models such as HarmBench typically output almost binary results (either ≈0 or ≈1), preventing more nuanced analysis.
>
> Please also see the general comment for data and more thorough discussion.
>
> [1] Souly, Alexandra, et al. "A StrongREJECT for empty jailbreaks." Advances in Neural Information Processing Systems 37 (2024): 125416-125440.
> ### W3 Sensitivity to parameters in Figure 8
> We have followed your suggestion by updating Figure 8 to include various thresholds, confirming that **our findings are robust across thresholds**.
> ### W4/Q2 On sampling temperature & comparing the entropy objective to high-temperature sampling
> We use $T=0.7$ for our experiments (now stated directly in Section 6.1).
> We conducted experiments comparing higher temperatures to the entropy objective, but found that at the temperatures necessary to produce harmful responses given our sampling budget, model responses often become incoherent. The entropy objective avoids this problem since it only increases the entropy of the first token, while allowing for coherent generation afterwards.
>
> New experiments in App. J show that pure generation with higher temperatures does not reach the same harmfulness as GCG with the entropy objective. For example, for Gemma 3, we achieve an ASR of $0.79@n=50$, but pure generation even with $n=1000$ is still significantly lower at 0.51. This finding is consistent across most other models too (Llama 2 7B DA 0.55@50 vs 0.09@1000, Llama 3.1 8B 0.84@50 vs 0.71@1000).
>
> Please also see the general comment for more details.
> |Temp|n=1|n=10|n=100|n=1000|
> |-|-|-|-|-|
> |0.0|0.0875|-|-|-|
> |0.7|0.0875|0.15|0.23|0.32|
> |1.0|0.0875|0.17|0.28|0.36|
> ### W5 Reproducibility of harmful responses
> Our threat model explicitly does not focus on reliable jailbreaks, instead investigating worst-case behavior. However, while any individual harmful response may be hard to reproduce, the aggregate trends are highly robust, meaning that at a given sample level, it is highly likely that at least some of the responses are harmful when reproduced. Finally, our experiments are fully reproducible due to a codebase designed for deterministic execution.
> ### W6 Why isn't there a yellow bar (250 samples) for optimize-then-sample in Figure 4?
> Excellent question! This is an artifact of the experimental design. As outlined in section 6.1, we collected $T\times n$ ($n=50$) responses to allow us to investigate different sampling schedules post-hoc. This setup does not allow us to report results for optimize-then-sample at n larger than 50. Re-running experiments for $n=250$ would be very computationally expensive.
> ### W7 Questions regarding the compute-optimal frontier
> Your observation is correct. The frontier shows the Pareto-optimal allocation for a *total* budget: Every point on the line corresponds to a different *total* budget and shows the optimal allocation for that particular budget. If the optimal ratio was constant at 1:100, the frontier would form a diagonal line through points with that ratio). Instead, we see that the optimal ratio is much higher, growing even further as the total FLOP budget increases, demonstrating sampling's efficiency advantage. We have also added a mathematical definition of the frontier to Appendix M.
> ### Q1 & Q2
> Please see W2 and W4.
>
> We hope these answers and modifications address your concerns.

---

> ### Comment · Reviewer_uZGG · 2025-11-26
>
> I've read other reviews and authors' responses. I will keep my score.

---

### Official Review · Reviewer_auPK · 2025-11-01

**Soundness:** 3
**Presentation:** 3
**Contribution:** 3
**Rating:** 6
**Confidence:** 5

**Summary:**

This paper proposes a perspective shift in the domain of LLM adversarial attacks. While most works on LLM safety only consider adversarial attacks in deterministic (T=0) settings, the authors suggest that we should focus more on the effects of random sampling (T>0) when discussing adversarial attacks and LLM safety. They propose a meta-algorithm for generating adversarial attacks in a sampling-aware manner that can be used to enhance any optimization-based attack algorithm. The authors show that their sample-aware optimization can drastically reduce the optimization costs of various SOTA attack algorithms, while also improving their efficacy. The authors also propose a new, label-free optimization objective based on first-token entropy, and show that sampling-aware optimization can be more effective when using this entropy-based loss.

**Strengths:**

The paper is well structured and easy to read. The paper motivation is clear, the main idea of using sampling for more exploration in the adversarial space at lower costs is simple yet effective.

The paper discusses a well-known, but not sufficiently highlighted issue in LLM safety evaluation: most works evaluate LLM safety only in the deterministic (T=0), greedy generation setting, which generally underestimates attack strength and overestimates model robustness. I think the paper does a good job at quantifying the effects of these shortcomings in evaluation protocols.

The claimed improvements in both efficiency and effectiveness are significant and worthy of community attention.

The paper presents thorough experiments on multiple settings, including a wide range of attacks, models and defences. The additional experiments, visualizations and insights are also interesting and valuable.

**Weaknesses:**

**Insufficient discussion on Temperature**

I think the temperature settings used in the experiments should be discussed in the main paper. While reading I was specifically looking for this information but it was only found in the appendix.

Moreover, I feel that temperature should be treated as a main hyper-parameter in the sampling algorithm as it could significantly affect its performance. The paper would be strengthened by including some ablations and discussions regarding the influence of temperature. I am interested in how the temperature could increase or decrease the attack success rate, the quality of harmful responses, the need for more or less sampling steps per iteration etc.

**Insufficient justification for the entropy objective**

While the idea of maximizing the entropy of the first token makes intuitive sense for trying to avoid safe completions, there is no theoretical justification for why optimizing for entropy would be better than optimizing for an affirmative completion. Moreover, while the results in Table 2 show some improvements, it is not very clear that the improvements are consistent across attack types or statistically significant. I think more evidence is needed to affirm that entropy maximization is actually better. For example the authors should analyze the distribution of harmfulness for completions generated by the affirmative and entropy prompts respectively. Finally, the claim that the responses for entropy-based prompts are more natural is purely subjective and only backed by 3 examples in Table 5. To support this claim the authors should provide a more complete analysis, including the differences in distribution for some similarity / naturality scores between the two approaches.

**Questions:**

1. The paper proposes improvements on adversarial attack generation techniques, but does not discuss any possible avenues for improving model robustness in light of their discoveries. While defence improvement can be considered outside the scope of this paper, I would like to hear the authors’ views on this issue.
2. The entropy maximization objective might have some other interesting effects on the model responses. Have the authors observed during their experiments responses that deviate from the original question, like random gibberish text, or answers to a different question?
3. Other questions that I have are just related to the weaknesses discussed above.

---

> ### Author Response · Authors · 2025-11-19
> **Answer to auPK**
>
> We thank the reviewer for their thorough review and are glad they find our contributions “significant and worthy of community attention”.
> We would now like to address their questions and concerns:
>
> ### W1 Regarding the sampling temperature.
> We agree that temperature is a key hyperparameter and now directly specify our decoding parameters (including temperature) in Section 6.1, in addition to Appendix J. We selected a temperature of 0.7 for most experiments to avoid overclaiming results.
> Preliminary experiments comparing sampling at different temperatures (summary results below) showed that **higher temperatures lead to even greater advantages for sampling** at a given sample count. However, **high-temperature samples are at higher risk of being incoherent and/or false positives**. Our setting was chosen because it already achieves strong performance with lower risk of incoherent generations and false positives.
> Please also see the general comment and App. J for a more thorough discussion of this topic.
>
> | Temp | n=1   | n=10  | n=100 |
> |------|-------|-------|-------|
> | 0.0  | 0.0875 | – | – |
> | 0.7  | 0.0875 | 0.15 | 0.23 |
> | 1.0  | 0.0875 | 0.17 | 0.28 |
>
> ### W2 Concerns about the entropy objective
> We appreciate the points raised and believe they may stem from a misunderstanding concerning our claims.
> The entropy objective and accompanying experiments are intended as a **proof-of-concept** illustrating how sampling-driven attack design can enable alternative paradigms. We **do not claim that it outperforms the affirmative objective overall**, only that it appears comparable, has interesting properties, and is in some settings (such as the T=5 setup) a bit better/easier to optimize. This has been clarified in the revised paper.
>
> In addition, we have **included new results in App. C, strengthening our claims regarding natural responses**:
>
> A **quantitative analysis** of the model output distribution over answers produced with the entropy and the affirmative objective, shows that responses elicited via the entropy objective have a **lower symmetric KL divergence with benign answers** (not just qualitatively).
> As suggested, we also provide **histograms for the distribution of harmfulness comparing the affirmative and entropy objective**.
>
> |                 | Symmetric KL |
> |----------------------|--------------|
> | Affirmative ⇔ Benign | 22.21        |
> | Entropy ⇔ Benign     | **17.68**        |
>
>
> ### Q1 Thoughts on defenses
> For us, the ultimate goal of attack development/red-teaming is to make models safer by discovering weaknesses so that we may fix them. In this context, we are particularly excited by the efficiency aspect of our work: in our view, the only paradigm that led to lasting improvements in model robustness in other domains like computer vision is adversarial training (AT) [2-5]. For LLMs, this is currently severely bottlenecked by the compute intensity of existing attacks, which prohibits efficient AT, and we are hopeful that our efficiency improvements can help make AT for LLMs more practical, thus leading to improved safety.
>
> [2] Bai, J., Wang, Y., Su, H., and Zhu, J. Recent Advances in Adversarial Training for Adversarial Robustness. International Joint Conference on Artificial Intelligence, 2021, pp. 4312–4321.\
> [3] Tramèr, F., Carlini, N., Brendel, W., and Madry, A. On Adaptive Attacks to Adversarial Example Defenses. Advances in Neural Information Processing Systems, 2020, pp. 1633–1645.\
> [4] Athalye, A., Carlini, N., and Wagner, D. Obfuscated Gradients Give a False Sense of Security: Circumventing Defenses to Adversarial Examples. International Conference on Machine Learning, 2018, pp. 274–283.\
> [5] Uesato, J., O’Donoghue, B., Kohli, P., and Oord, A. Adversarial Risk and the Dangers of Evaluating Against Weak Attacks. International Conference on Machine Learning, 2018, pp. 5025–5034.
>
> ### Q2 The entropy maximization objective might have interesting effects on the model responses. Have the authors observed during their experiments responses that deviate from the original question, like random gibberish text, or answers to a different question?
> The most typical “odd” responses we see are answers in another language. Interestingly, the vast majority of samples remain on topic, either answering or refusing the harmful prompt. Given the interest, we will responsibly release our dataset of samples to the research community.
> In response to other reviews, we have added histograms for the harmfulness distribution when using the entropy objective to Figure 10 in Appendix C, which you may also find interesting.
>
> Please let us know if these changes address your concerns.

---

> ### Comment · Reviewer_auPK · 2025-11-24
>
> Thank you for the detailed rebuttal! My concerns and questions have moslty been addressed. I really like figures 9 and 10, but figure 10 is a bit hard to interpret, particularly because the y-scales are different in each plot. Please consider using the same scale at least for each pair of plots corresponding to the same optimization step, as the conclusions might be slightly different. Also, I like that the authors provide some results with varying generation temperatures, but I think the paper would still benefit from a more detailed discussion on temperature (at least considering some more intermediate values). A plot in the style of fig 10 for the harmfullness distribution at different temperatures would be interesting to see (maybe also towards really high T values where the model starts to output nonsense).
>
> Taking into account the rebuttal discussions, I am leaning towards increasing my score to 8: Accept.

---

> > ### Author Response · Authors · 2025-11-26
> > **Reply to auPK**
> >
> > Thank you for your response!
> >
> > We have updated Figure 10 with consistent y-axis as suggested.\
> > In addition, we have run experiments using a range of temperatures using Llama 3.1 8B. Results are shown in Figure 35 in App. J (p.24).\
> > We can see that the achieved harm $\mathcal{H}_q$ peaks near temperature 1.0 for this model, making the main results of our paper (using a temperature of 0.7) rather conservative and suggesting that tweaking the temperature could yield even larger performance gains for sampling. At temperatures above 1.0, generations deteriorate and begin to become unstable.\
> > For the camera-ready, we will run these experiments for the other models from the paper as well and will also move the finalized figure and associated discussion into the main body of the paper (likely as part of Section 6.3).
> >
> > We hope our changes have addressed all your concerns and are happy that you are willing to increase your score.
> > Please let us know if you have any additional comments.

---

### Author Response · Authors · 2025-11-19
**General Comment**

We thank all reviewers for their helpful reviews and are encouraged that they find our work as bringing a "fresh perspective" (gbXv) that is "significant and worthy of community attention" (auPK). We are pleased that reviewers appreciated the novel model-agnostic, label-free entropy objective (PvN7, gbXv, uZGG) and the "thorough experiments" (auPK).

We would now like to address the two main shared concerns, which revolve around temperature as a hyperparameter for sampling (auPK, uZGG, PvN7) and the reliance on StrongREJECT as a judge model (uZGG, gbXv, PvN7).

### Temperature as a hyperparameter (auPK, uZGG, PvN7).
Reviewers noted that temperature and other important sampling parameters should be prominently presented in the main body of the paper. We agree that decoding parameters are key hyperparameters and updated Section 6.1 and Appendix L to explicitly describe our settings (multinomial sampling with temperature 0.7).

In addition, reviewers were curious about sampling's performance at different temperature settings and our rationale for the chosen threshold. To address these questions, we now **include results for sampling with different temperatures** (excerpt below, full results & discussion in Appendix J).

Our experiments comparing sampling at different temperatures show that **higher temperatures lead to even greater advantages for sampling** at a given sample count. However, we found **high-temperature generations to be at higher risk of being incoherent or false positives** rather than genuinely harmful. As a result, we chose 0.7 as performance was already strong, and to avoid incoherent outputs with higher false positive rates.

| ASR @ Temp | n=1   | n=10  | n=100 |
|------|-------|-------|-------|
| 0.0  | 0.0875 | – | – |
| 0.7  | 0.0875 | 0.15 | 0.23 |
| 1.0  | 0.0875 | 0.17 | 0.28 |


### Focus on StrongREJECT (uZGG, gbXv, PvN7)
To address concerns about automated judging, we conducted a **gold-standard human study** (details in Appendix K) and found that the **conclusions drawn from StrongREJECT remain unchanged** when evaluated with human judgments. For the human study, two paid annotators followed the instructions from p.25 of [1] and were blinded to the model, attack, and generation parameters.
The added results show that correcting for false positives of StrongREJECT does **not alter the overall findings**.
All key findings such as the effectiveness of sampling-aware attacks and the relative ranking of models, **persist in the human-labeled scenario,** showing that the findings are robust and not sensitive to a particular classifier being used.
We also find that, perhaps counterintuitively, ASR$_q@50$ is *less* influenced by such corrections than ASR$_q@1$ because most successful attacks contain multiple harmful completions for each prompt, making it unlikely that all are false positives.

| Metric        | AutoDAN | BEAST  | PAIR   | REINFORCE-GCG | GCG    |
|---------------|---------|--------|--------|----------------|--------|
| ASR$_q@1$        | 0.34   | 0.08 | 0.08 | 0.23          | 0.17  |
| ASR$_q@1$ human  | 0.27  | 0.06 | 0.06 | 0.18          | 0.13   |
| ASR$_q@50$       | 0.51    | 0.22  | 0.24 | 0.55         | 0.47 |
| ASR$_q@50$ human | 0.50  | 0.21   | 0.23 | 0.53          | 0.45  |
| Δ$_{1\rightarrow50}$  | 0.18   | 0.14 | 0.16   | 0.32         | 0.30 |
| Δ$_{1\rightarrow50}$ human    | **0.24**   | **0.15** | **0.17**  | **0.35**           | **0.32**  |

We would also like to provide more context on our rationale for choosing StrongREJECT over alternatives.

In our evaluations of open-source judge models at the start of the project, StrongREJECT was the strictest judge and aligned most closely with human judgment. In contrast, alternatives such as HarmBench tended to produce significantly more false positives and near-binary outputs that prevented more nuanced analysis. We present results from HarmBench, Llama Guard 3, and StrongREJECT below, showing that StrongREJECT is much stricter than the alternatives.

|                       | HarmBench neg | HarmBench pos |
|-----------------------|---------------|---------------|
| Strong Reject pos     | 55            | 946           |
| Strong Reject neg     | 2226          | 773           |

|                       | LlamaGuard 3 neg | LlamaGuard 3 pos |
|-----------------------|-------------------|-------------------|
| Strong Reject pos     | 146               | 855               |
| Strong Reject neg     | 1888              | 1111              |


[1] Souly, Alexandra, et al. "A StrongREJECT for empty jailbreaks." Advances in Neural Information Processing Systems 37 (2024): 125416-125440.

---

### Author Response · Authors · 2025-12-03
**Final summary**

We again thank all reviewers for their constructive engagement and are encouraged by the consistent recognition of our contribution and perspective offered by our work. Reviewers highlighted the **fresh viewpoint** the paper introduces (gbXv), its **significance for the community** (auPK), the **novel model-agnostic and label-free entropy objective** (PvN7, gbXv, uZGG), and the **thorough experimental analysis** (auPK).

The discussion involved two shared themes: the **role of sampling temperature** and the **reliance on StrongREJECT** for measuring harmfulness. Both points are now addressed comprehensively:
1. All decoding parameters, are specified directly in the main paper, and **extensive ablations covering a wide range of temperatures** were added. (auPK, gbXv, uZGG)
2. A **gold-standard human study** was conducted, confirming that **all central conclusions hold when evaluated by humans** rather than an automated judge. Additional comparisons with alternative judges support our choice of StrongREJECT. (gbXv, uZGG, PvN7)

All remaining comments are resolved through new analyses and clarifications, including:
- **experiments on larger models**, including Llama 3.1 70B (gbXv, PvN7)
- updated figures showing our **results hold across multiple harm thresholds** (uZGG, auPK)
- supporting ablations and **expanded rationale for the entropy objective** (PvN7)
- a clear, formal definition and explanation of the compute-optimal frontier (uZGG)
- clarification of the used sampling schedules and their trade-offs (PvN7, gbXv)

All updates are highlighted in blue in the revised manuscript.
We again thank all reviews for their helpful comments, which led to refinements that meaningfully strengthen our work, reinforcing the robustness of the main results and even yielded interesting new findings (e.g., regarding the optimal sampling temperature). We believe our work has the potential to contribute to more reliable LLM evaluations in adversarial contexts and, ultimately, to safer LLMs.

---

### Meta-Review · Area_Chair_pGiY · 2026-01-13

**Summary:**

This paper argues that greedy decoding underestimates jailbreak risk. Then, the authors proposed sampling-aware attacks that treat repeated sampling as part of the adversary’s budget. The results reported large gains in attack success and efficiency. This paper also proposed a label-free first-token entropy objective as a proof-of-concept. There were three borderline positive reviews and one negative reviews. One of the positive reviewer acknowledge the rebuttal and mentioned about leaning clear accept. After reading all rebuttal, the AC thinks that the strengths of the paper are clear and the common issues are well addressed in the rebuttal. The paper can be accepted.

**Reviewer Concerns:**

- **Decoding parameters/temperature**. The authors added decoding parameters in main text, temperature ablations, reframed entropy as proof-of-concept, and added distributional analysis. The AC thinks this can addressed the corresponding issues.
- **Reliance on StrongREJECT**. The human study was added.
- **Larger models**. The 70B experiments were added.
- **More comparisons**. The authors discussed and answered this question.

**Reviewer Scores:**

After the reading rebuttal, it is clear to the AC that there are three positive scores. For the only negative score, the AC thinks there were no significant issues proposed and the authors were able to address them during the rebuttal.

---

### Decision · Program_Chairs · 2026-01-26

Accept (Poster)